# Safety and Feasibility of Catheter Ablation Procedures in Patients with Bleeding Disorders

**DOI:** 10.3390/jcm11236956

**Published:** 2022-11-25

**Authors:** Marcel Feher, Ardan M. Saguner, Bettina Kirstein, Julia Vogler, Charlotte Eitel, Huong-Lan Phan, Ahmad Keelani, Tolga Cimen, Sascha Hatahet, Darko Trajanoski, Omar Samara, Karl-Heinz Kuck, Roland R. Tilz, Christian-H. Heeger

**Affiliations:** 1Department of Rhythmology, University Heart Center Lübeck, University Hospital Schleswig-Holstein, 24105 Kiel, Germany; 2Department of Cardiology, Universitätsspital Zurich, 8091 Zurich, Switzerland; 3LANS Cardio, 20354 Hamburg, Germany; 4German Center for Cardiovascular Research (DZHK), Partner Site Lübeck, 20246 Hamburg, Germany

**Keywords:** catheter ablation, pulmonary vein isolation, bleeding disorders

## Abstract

Aims/Objectives: Patients with bleeding disorders are a rare and complex population in catheter ablation (CA) procedures. The most common types of bleeding disorders are von Willebrand disease (VWD) and hemophilia A (HA). Patients with VWD or HA tend to have a higher risk of bleeding complications compared to other patients. There is a lack of data concerning peri- and postinterventional coagulation treatment. We sought to assess the optimal management of patients with VWD and HA referred for catheter ablation procedures. Methods and Results: In this study, we analyzed patients with VWD or HA undergoing CA procedures at two centers in Germany and Switzerland between 2016 and 2021. Clotting factors were administered in conjunction with hemostaseological recommendations. CA was performed as per the institutional standard. During the procedure, unfractionated heparin (UFH) was given intravenously with respect to the activated clotting time (ACT). Primary endpoints included the feasibility of the procedure, bleeding complications, and thromboembolic events during the procedure. Secondary endpoints included bleeding complications and thromboembolic events up to one year after catheter ablation. A total of seven patients (three VWD Type I, one VWD Type IIa, three HA) underwent 10 catheter ablation procedures (pulmonary vein isolation (PVI): two × radiofrequency (RF), one × laser balloon (LB), one × cryoballoon (CB); PVI + cavotricuspid isthmus (CTI): one × RF; PVI + left atrial appendage isolation (LAAI): one × RF; Premature ventricular contraction (PVC): three × RF; Atrioventricular nodal reentrant tachycardia (AVNRT): one × RF). VWD patients received 2000–3000 IE Wilate i.v. 30 to 45 min prior to ablation. Patients with HA received 2000–3000 IE factor VIII before the procedure. All patients undergoing PVI received UFH (cumulative dose 9000–18,000 IE) with a target ACT of >300 s. All patients after PVI were started on oral anticoagulation (OAC) 12 h after ablation. Two patients received aspirin (acetylsalicylic acid; ASA) for 4 weeks after the ablation of left-sided PVCs. No anticoagulation was prescribed after slow pathway modulation in a case with AVNRT. No bleeding complications or thromboembolic events were reported. During a follow-up of one year, one case of gastrointestinal bleeding occurred following OAC withdrawal after LAA occlusion. Conclusions: After the substitution of clotting factors, catheter ablation in patients with VWD and HA seems to be safe and feasible.

## 1. Introduction

Von Willebrand disease (VWD) and hemophilia A (HA) are the most common types of bleeding disorders. Many patients with VWD are not aware of the disease because signs are mild or absent. In most patients, VWD is inherited from one or both parents and caused by an abnormal gene that controls the von Willebrand factor, which is a protein that plays a key role in blood clotting. The prevalence of symptomatic VWD is approximately 1 in 1000 (0.1%) [1]. There are three main types of VWD: in VWD type I, von Willebrand factor (VWF) is deficient, concerning approximately 70–80% of the patients. In type II, there is a quantitative deficiency with low VWF activity levels in 15–20%. A subentity is the so-called acquired VWD that is frequently associated with monoclonal gammopathies, lymphoproliferative, myeloproliferative, and autoimmune disorders. In VWD Type III there is practically no functional VWF (<5%) in the serum, with the highest bleeding risk [2].

HA is an X-linked recessive, congenital disorder with a lack of factor VIII leading to spontaneous bleeding [3]. Mostly prevalent in male births, it occurs in approximately 1 in 5000 (0.02%) [4]. HA can be divided into three forms according to residual factor VIII activity: severe (FVIII_a_ < 1%, <1 IU/dl), moderate (FVIII_a_ 1–5%, 1–5 IU/dl), or mild (FVIII_a_ < 5–40%, 6 IU/dl to <40 IU/dl) [3,5].

By now, life expectancy in both bleeding disorders is almost normal with an increased incidence of cardiovascular diseases and cardiac arrhythmias [6].

Patients with these disorders are still a rare and complex population in catheter ablation (CA) procedures. They tend to have a higher risk of bleeding complications compared to other patients. However, they also have a higher risk of thrombosis. The optimal anticoagulation management is still questionable.

In CA procedures, anticoagulants (mainly heparin) are given periprocedurally, increasing the bleeding risk in this population even more. There is no valuable data on how to estimate the risk of ischemic stroke. Common risk scores such as the CHA_2_DS_2_-VASc score and HAS-BLED score are not validated properly in VWD and HA, overestimating the thromboembolic risk [7]. In the ESC practical guidelines on “Novel Oral Anticoagulants for Atrial Fibrillation”, there are no recommendations for patients with bleeding disorders referred for CA [8].

In this report, we describe our experiences with patients with bleeding disorders undergoing any type of CA.

## 2. Materials and Methods

In this multi-center study, we analyzed patients with bleeding disorders referred for CA. Data were acquired from the University of Luebeck, Germany, and the University Hospital of Zurich, Switzerland, from 2016 to 2021. Patients with paroxysmal or persistent atrial fibrillation ablation (PAF and PersAF), PVC ablation, and slow pathway ablation due to AVNRT were included. Baseline characteristics, comorbidities, type of bleeding disorder, type of arrhythmia, OAC as well as peri- and postprocedural outcomes were extracted from medical data files. Each center was responsible for correct ethical standards and regulations. The registry was approved by the local ethical review board (Lübeck ablation registry ethical review board number: WF-028/15) and all participants provided written informed consent. All investigations were performed in compliance with the ethical standards laid down in the 1964 Declaration of Helsinki and its later amendments.

### 2.1. Pulmonary Vein Isolation

Prior to PVI, LA and LAA thrombi were excluded by transesophageal echocardiography. All ablations were performed under intravenous sedation using propofol following the recommendations of a position paper by the German Cardiac Society [Tilz2017]. For catheter access, 2–3 sheaths (each 8.5 Fr) were inserted into the femoral veins after an ultrasound-guided puncture. Multipolar diagnostic catheters were placed as per the standard of care. Single or double transseptal access and the insertion of transseptal sheaths (TS) were followed by catheter ablation as per the institutional standard. PVI was achieved either by a single-shot device (cryoballoon (CB) or laser balloon (LB)) or by radiofrequency (RF) ablation. In RF ablation a three-dimensional cardiac mapping system (Carto, Biosense Webster, Johnson & Johnson, Diamond Barr, CA, USA, Figure 1) was used for the three-dimensional reconstruction of the LA anatomy. UFH was administered targeting an ACT of >300 s.

After the sheaths were removed, hemostasis was achieved by manual compression. A pressure bandage was applied for 4 to 6 h and the patient was required to lie on their back. When there was no painful swelling requiring drug medication, superficial bleeding, or bruising of the tissue, the patient was eligible to stand up and walk.

### 2.2. PVC Ablation

The ablation of frequent PVC was performed in accordance with our institutional protocol. In addition to 2–3 venous sheaths, one arterial line (6 Fr) was placed next to the femoral sheaths. In the case of PVCs of the left ventricle (LV), TS was performed as described before. During PVC ablation of the LV, the target ACT was >300 s, reached by i.v. UFH. In PVC ablation of the right ventricle, 3000 IE UFH was administered after a femoral groin puncture. At the end of the procedure, the femoral arterial access site was closed by the insertion of a vascular closure system (Angio-Seal^TM^ VIP Vascular Closure Device, Terumo, Somerset, NJ, USA. Further management of the puncture site was carried out as described before.

### 2.3. AVNRT Ablation

AVNRT ablation and periprocedural anticoagulation management were performed in accordance with the institutional protocol. The management of the femoral access site was carried out as described before.

### 2.4. Coagulation Management

In accordance with the hemostaseological recommendations, blood levels were taken on the day of admission. Blood samples included baseline factor VIII levels in all cases. Additionally, VWF antigen and VWF activity levels were assessed in VWD. If the patient was on OAC, we withdrew the OAC 36 h prior to the procedure.

#### 2.4.1. Pre- and Intraprocedural Coagulation Management

Clotting factor correction was given intravenously 30 to 45 min prior to puncture of the groin. In VWD, 20–30 IE/kg KG Wilate (Coagulation factor VIII/VWF complex) was administered according to factor VIII, VWF antigen, and VWF activity levels. In HA 20–30 IE kg/KG, Haemoctin or NovoEight (Coagulation factor VIII) was given if factor VIII levels were <60%. During CA, UFH was given i.v. with a target ACT of 300 to 350 s in accordance with local ablation protocols.

#### 2.4.2. Postprocedural Management

After PVI, OAC or low molecular weight heparin was given 12 h after CA. Tranexamic acid was given for 3 to 5 days in respect of the hemostaseological consultation. Anticoagulation was continued for at least 3 months and according to the CHA_2_DS_2_-VASc score afterward.

After PVC ablation of the LV, ASA was continued for 4 weeks. In AVNRT, no anticoagulants were prescribed.

### 2.5. Endpoints

Primary endpoints included the feasibility of the procedure, bleeding complications, and thromboembolic events during the procedure. Secondary endpoints included bleeding complications, thromboembolic events, and freedom from arrhythmia up to one year after catheter ablation.

## 3. Results

A total of seven patients (mean age 51 years, five male) underwent 10 catheter ablation procedures (Table 1). Three patients had VWD Type I, and one had VWD Type IIa. Three patients had mild HA.

The median CHA_2_DS_2_-VASc score was 1 (IQR 0–3), and the median HAS-BLED score was 1 (IQR 1–2). At baseline, only one patient was on oral anticoagulation (patient #4 was on dabigatran 110 mg bid).

Prior to ablation, a median dose of 2000 IE factor VIII was given intravenously in HA. In one patient (#3 with PVC) no clotting correction was necessary.

During PVI, UFH was given to maintain an ACT > 300 s. In PVI cases, the median heparin dose was 14.500 IE (IQR 12,000–18,000). In PVI (plus redo procedures including LAA isolation), the median procedure time was 98.2 min (IQR 70–125). The median procedure time of non-PVI cases was 95.5 min (IQR 80–111). In RF PVC ablation, the median heparin dose was 8000 IE (IQR 3000–14,000). One patient received a cumulative dose of 9000 IE heparin during an electrophysiological study, excluding a left-sided accessory pathway. The diagnosis of AVNRT was confirmed during the procedure with successful ablation of the slow pathway requiring 3 RF applications.

Oral anticoagulation was prescribed in all PVI cases 12 h after ablation. Three patients received DOAC (one rivaroxaban, one apixaban, and one dabigatran), and another patient, warfarin (Table 2). Warfarin was withdrawn from patient #2, 3 months after LB PVI ablation. In patients #1 and #4, DOAC was withdrawn 4 months after the last PVI and 1 month after LAAC implantation, respectively. Two patients received ASA for 4 weeks after ablation for premature ventricular contractions of the left ventricle. One of these patients received 1500 IE NovoEight three times a week during the ingestion of ASA. No anticoagulation was prescribed after slow pathway ablation in any patient with AVNRT. Tranexamic acid (3 × 1 g/d) was given in two cases for 3 to 5 days in accordance with hemostaseological recommendations. In one case with HA, 1000 IE Haemoctin was administered for 3 d.

In the follow-up (Table 3), patient #1 had two redo procedures after a PersAF recurrence following CTI and consecutive LAA isolation. Patient #4 had one redo procedure without any complications.

In this complex patient population, no bleeding complications or thromboembolic events were reported. Follow-up at one year showed gastrointestinal bleeding (angiodysplasia) in one case following OAC withdrawal 3 months after LAA occlusion.

## 4. Discussion

In this prospective cohort study, we sought to investigate the safety and feasibility of patients with bleeding disorders undergoing catheter ablation procedures.

The procedural outcomes of patients undergoing PVI were comparable to the general population. Patients with PersAF required one to two redo procedures. Only one patient with PAF had no recurrence of atrial arrhythmias. In the literature, success rates of up to 75% with PAF were described, whereas, with PersAF, the success rates are only around 50%. The single procedure success rate after PVI (33%) was lower in our cohort. Two out of three patients had PersAF. After one redo procedure, the success rate increased to 66.7%. In one case, a second redo procedure was necessary to achieve sinus rhythm in the follow-up [9].

In all cases, there were no periprocedural major bleeding complications. According to the ESC Guidelines for AF, the incidence of major groin complications in the general population is about 0.9% [9]. Groin hematomas can especially lead to discomfort and a lengthened hospital stay. Van der Valk et al. reported a higher incidence of groin hematomas in patients with bleeding disorders (25%) [10], however, no major complications occurred in this study. A puncture of the femoral vein was performed under ultrasound guidance. Other studies showed a reduction in complications using an ultrasound device [11], and our in-house rate of groin hematomas is about 5%.

Almost all AF patients received a DOAC 8–12 h after the procedure instead of a vitamin K antagonist. There was no need for bridging with LMWH, except in one case when DOACs were not common practice. A DOAC reduces the risk of thromboembolic events and bleeding, as previous studies have shown [12]. The use of apixaban, dabigatran and rivaroxaban were safe. Any DOAC can be used for anticoagulation in the general population. In the cases of major bleeding, there are antidotes for all common DOACs [13].

Within 36 h before any CA, DOACs were withdrawn and deemed to be safe. The period of time that DOACs should be prescribed for in this patient population is unclear. Common risk scores such as the CHA_2_DS_2_-VASc score may overestimate the embolic stroke risk in patients with bleeding disorders [7,14]. In our case series, we used this risk score to estimate how long DOACs should be given and prescribed in accordance with the general recommendations. Data from Schutgens et al. postulate starting long-term anticoagulation only in patients with a CHA_2_DS_2_-VASc score ≥2 if the baseline FVIII levels are 20% in patients with HA. If FVIII levels are below a trough of 20%, DOACs can be considered in patients with a CHA_2_DS_2_-VASc score ≥4 [14]. In FVIII levels <20%, other options such as antiplatelet therapy with low-dose ASA can be discussed, although the role in stroke prevention in AF is still questionable [9].

In the future we will adapt our approach to reduce the bleeding risk for those patients even further, considering these cutoffs before starting any long-term anticoagulation [15].

The substitution of FVIII and VWF was given in accordance with hemostaseological recommendations. The HA guidelines recommend a preoperative peak factor level of 60–80% as a low-dose practice pattern for major surgeries [16,17].

In our experience, these recommendations for preoperative replacement therapy were safe with any procedural-associated complications. The laboratory results in the follow-up were unfortunately missing.

In one patient with known angiodysplasia under DOAC, dabigatran gastrointestinal bleeding occurred. There was no need to use an antidote. The DOAC was withdrawn and two bags of red blood cells were given. In this case, we decided to implant an LAAC to withdraw the DOAC afterward. To our knowledge, there is little data on the implantation of LAAC in patients with bleeding disorders [18]. In one more patient (#1), the implantation of an LAAC after LAA isolation was safe. As procedural safety increases, LAAC may be a reasonable option to withdraw long-standing DOAC therapy in this population [19].

During follow-up, patient #4, with an acquired VWD, had stable disease, receiving bortezomib. In very few case reports, bortezomib was effective in reducing the bleeding risk in acquired VWD and should be taken into consideration in selected cases [20,21].

The use of antiplatelet agents in two patients after PVC ablation for 4 weeks was safe. No bleeding complications occurred during antiplatelet therapy for this short period of time. There are no data concerning the short-term safety of antiplatelet therapy. For the long-term use of antithrombotic agents, a French registry reported increased bleeding rates in 68 patients with HA [22].

Studies showed bleeding complications, especially in the first days after CA [10]. In three patients with VWD or HA, Tranexamic acid (TA) was given for 3 to 5 days after the procedure. Tranexamic acid is an antifibrinolytic agent which can reduce bleeding by inhibiting hyperfibrinolysis. Because TA has nearly no side effects, it can be used after invasive procedures to further reduce the bleeding risk [23,24].

## 5. Limitations

This study has some limitations. Due to the small number of patients, there was no statistical significance. Therefore, a descriptive study presentation was conducted. It would be desirable to observe FVIII/VWF levels in the follow-up, even though there were no severe complications. In the lack of general recommendations concerning patients with bleeding disorders undergoing catheter ablation procedures, a comparison of other studies may over- or underestimate complications, for example, as seen with groin complications.

## 6. Conclusions

Following the substitution of clotting factors according to the blood levels before the procedure, catheter ablation in patients with VWD and HA seems to be feasible and safe.

## Figures and Tables

**Figure 1 jcm-11-06956-f001:**
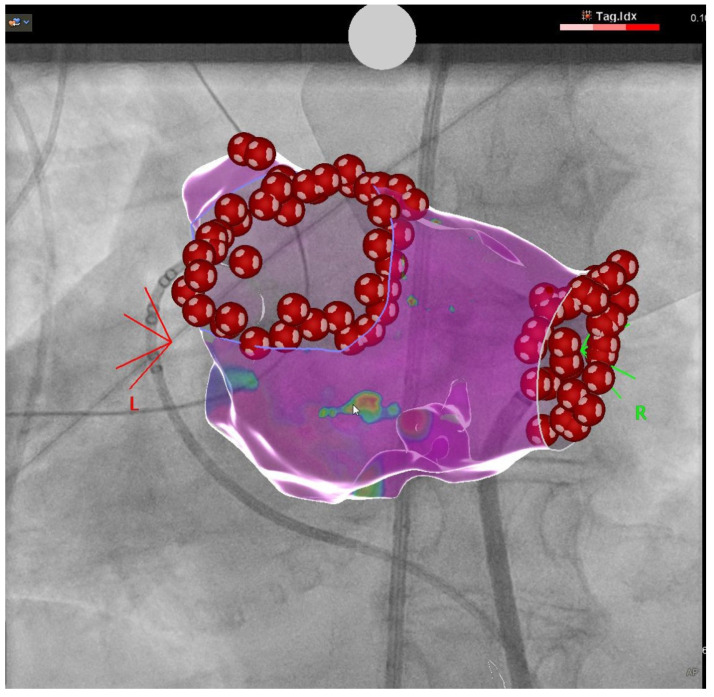
Electroanatomical reconstruction of a left atrium utilizing a 3D mapping system. PA view.

**Table 1 jcm-11-06956-t001:** Baseline characteristics.

Pt	Age	BMI	Bleeding Disorder	Clotting Levels [%]	Arrhythmia	CHA_2_DS_2_-VASc Score (HAS-BLED) Score	OAC Baseline	LV-EF
								[%]
1	57	22.6	VWD I	FVIII: 72 vWF_Ac_: 64 Ac/Ag: 0.82 *	PersAF	3 (2)	None	55
2	56	25.7	HA	FVIII: 15	PAF	1 (1)	None	55
3	23	21.5	VWD I	FVIII: 73 vWF_Ac_: 53 Ac/Ag: 0.92	PVC	0 (1)	None	55
4	76	31.6	VWD IIA	FVIII: 64 vWF_Ac_: 37 Ac: 37%	PersAF	4 (4)	Dabigatran	40–45
5	27	28.4	HA	FVIII: 22	PVC	1 (1)	None	55
6	45	22.2	VWD I	FVIII: 58 vWF_Ac_: 37 Ag/Ac: 0.79	AVNRT	0 (0)	None	55
7	72	20.7	HA	FVIII: 19	PVC	1 (2)	None	63%

Abbreviations: von Willebrand disease (VWD), hemophilia A (HA), factor VIII (FVIII), von Willebrand factor activity (VWF_Ac_), oral anticoagulation (OAC), left-ventricular ejection fraction (LV-EF), body mass index (BMI), premature ventricular contractions (PVC), atrioventricular nodal reentry tachycardia (AVNRT), paroxysmal atrial fibrillation (PAF), persistent atrial fibrillation (PersAF); *: cutoff Ac/Ag > 0.7.

**Table 2 jcm-11-06956-t002:** Pre-, peri- and postprocedural anticoagulation management.

Pt	Procedure	Procedure Time	Preprocedural Management	Periprocedural Management	Postprocedural Management	Complications
1	Re-PVI	125 min	2000 IE Wilate	10,000 IE Heparin	Tranexam acid 3 × 1 g/3 d Rivaroxaban	-
	Re-re-PVI	95 min	2000 IE Wilate	12,000 IE Heparin	Apixaban	-
	CTI + LAAI	145 min	2000 IE Wilate	12,000 IE Heparin	Apixaban	
2	Laser-PVI	50 min	2000 IE Haemoctin	14,000 IE Heparin	Warfarin	-
3	RF PVC	120 min	None (FVIII > 60%)	8000 IE Heparin	-	-
4	Cryo-PVI	70 min	3000 IE Wilate	18,000 IE Heparin	Dabigatran	-
	RF-PVI	104 min	3000 IE Wilate	21,000 IE Heparin	Dabigatran	-
5	RF PVC	70 min	2000 IE Haemoctin	14,000 IE Heparin	1000 IE Haemoctin/3 d 4 week ASA	-
6	RF AVNRT	91 min	2000 IE Wilate	9000 IE Heparin *	Tranexam acid 3 × 1 g/5 d 1000 IE Wilate/2 d	-
7	RF PVC	100 min	3000 IE NovoEight	3000 IE Heparin	1500 IE NovoEight 12 h and 24 h after the procedure 1500 IE NovoEight 3×/wk until ASA withdrawal; 4 week ASA	Minor hematoma, no Hb drop
		Median PVI: 98.2 (IQR 70–125) Median other: 95.5 (IQR 80–110)	Median Wilate: 2300 (IQR 2–3) Median Factor VIII: 2000 (IQR 2–3)	Median Heparin: 14,500 (IQR 12,000–18,000) Median Heparin 8000 (IQR 3000–14,000)		

Abbreviations: pulmonary vein isolation (PVI), oral anticoagulation (OAC), radiofrequency (RF), cavotricuspid isthmus (CTI), and left atrial appendage isolation (LAAI); *: diagnostic ep study of the left atrium + confirmation and ablation of AVNRT.

**Table 3 jcm-11-06956-t003:** Follow-up.

Pt	Procedure	DOP	Anticoagulation	OAC/APT Withdrawal	FFA	Complications	Further Interventions
1	Redo-PVI	08/2017	Rivaroxaban	-	-	-	
	Redo-PVI	05/2018	Apixaban	-	-	-	
	CTI + LAAI	02/2019	Apixaban	05/2019	02/2019	-	LAAC 04/2019 after LAAI
2	Laser-PVI	05/2018	Warfarin	08/2018	05/2018	-	
3	RF PVC	12/2019	-	-	12/2019	-	
4	Cryo-PVI	10/2020	Dabigatran	-	-	GI-Bleeding	
	RF-PVI	02/2021	Dabigatran	05/2021	12/2021	-	LAAC 04/2021 after GI-Bleeding
5	RF PVC	03/2021	ASA	04/2021	03/2021	-	
6	RF AVNRT	08/2019	-	-	08/2019	-	
7	RF PVC	08/2021	ASA	09/2021	08/2021	-	

Abbreviations: antiplatelet therapy (APT), patient (Pt), pulmonary vein isolation (PVI), oral anticoagulation (OAC), radiofrequency (RF), cavotricuspid isthmus (CTI), left atrial appendage isolation (LAAI), date of procedure (DOP), and freedom from arrhythmia (FFA).

## Data Availability

Non-digital data supporting this study are curated at the Study Center of the Department of Rhythmology, University Hospital Schleswig-Holstein, Germany.

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
