# Peer review of "Safety and Feasibility of Catheter Ablation Procedures in Patients with Bleeding Disorders"

_jcm, 2022, doi:10.3390/jcm11236956_

Round 1

Reviewer 1 Report

This study analyzed 7 patients with bleeding disorders of von willebrand disease or hemophilia A, who undergoing catheter ablation at two centers between 2016 to 2021. The study is well-written and the result is demonstrated clearly.

Major comments.

Due to the small patient numbers of the study, this study could be demonstrated as the case series, but the procedure safety, risk of thromboemtolic event and complications are hard to conclude and analyzed statistically.

Discussion:

Current AF guideline suggest uninterrupted(just stop OAC on the day of catheter ablation and take OAC on the same day after hemostatis) or minimally interrupted OAC use , do authours suggest to stop the DOAC 36 hours before catheter ablation in patients with bleeding disorders? 

 It is quite interesting that the previous study demonstrated DOAC can be considered in          

patients with a CHA2DS2-Vasc-Score ≥ 4 for patients with Factor VIII levels below trough of 20%. Did authors apply it to current cohort patients? 

Minor comment

1. line 68: typo "und" HASBLED-score 

"and" HASBLED-score

2. line 105:  lack of comma in the sentence:

In addition to 2 – 3 venous sheaths one arterial line (6 Fr) was placed next to the femoral sheaths. 

In addition to 2 – 3 venous sheaths, one arterial line (6 Fr) was placed next to the femoral sheaths. 

3. Results:

line 142: how many patients were male gender?

 A total  of  7 patients (mean age 51 years, xx male)  underwent  10 catheter  ablation 

4. How many HA patients were enrooled in the study? two or three?

line 143-144: two patients had mild HA. Theere were 3 patients with HA demonstrated in Table 1

5. Grammer

Puncture of the femoral vein was performed under ultrasound "guiding".

Puncture of the femoral vein was performed under ultrasound "guidance".

Author Response

The authors thank the reviewer for the valuable comments. We revised the paper according to the reviewers suggestions.

This study analyzed 7 patients with bleeding disorders of von willebrand disease or hemophilia A, who undergoing catheter ablation at two centers between 2016 to 2021. The study is well-written and the result is demonstrated clearly.

Major comments.

Due to the small patient numbers of the study, this study could be demonstrated as the case series, but the procedure safety, risk of thromboemtolic event and complications are hard to conclude and analyzed statistically.

We agree with the reviewers statement. Indeed it is very hard to find patients with vWD and catheter ablation procedures. We asked also other EP centers, yet in many centers those patients wont get treated invasively at all. Therefore we think that our small study is of great interest for other centers.

The conclusion is carefully suggesting that patients with vWD can be treated by catheter ablation after consultation of an bleeding disorders specialist.

Discussion:

Current AF guideline suggest uninterrupted(just stop OAC on the day of catheter ablation and take OAC on the same day after hemostatis) or minimally interrupted OAC use , do authours suggest to stop the DOAC 36 hours before catheter ablation in patients with bleeding disorders? 

If the patient was on OAC we stopped OAC 36h before ablation.

Methods:

2.4. Coagulation management

In accordance to the hemostaseological recommendations, blood levels were taken on the day of admission. Blood samples included baseline Factor VIII levels in all cases. Additionally, VWF antigen and VWF activity levels were assessed in VWD. If the patient was on OAC, we withdrew the OAC 36h prior to the procedure.

It is quite interesting that the previous study demonstrated DOAC can be considered in patients with a CHA2DS2-Vasc-Score ≥ 4 for patients with Factor VIII levels below trough of 20%. Did authors apply it to current cohort patients? 

No, we always followed the hemostaseological recommendations.

Minor comment

1. line 68: typo "und" HASBLED-score 

"and" HASBLED-score

Thank you. We revised this typo.

2. line 105:  lack of comma in the sentence:

In addition to 2 – 3 venous sheaths one arterial line (6 Fr) was placed next to the femoral sheaths. 

In addition to 2 – 3 venous sheaths, one arterial line (6 Fr) was placed next to the femoral sheaths. 

Thank you. We revised this typo.

3. Results:

line 142: how many patients were male gender?

 A total  of  7 patients (mean age 51 years, xx male)  underwent  10 catheter  ablation.

Thank you very much. We revised the paper

A total  of  7 patients (mean age 51 years, 5 male)  underwent  10 catheter ablation.

4. How many HA patients were enrooled in the study? two or three?

line 143-144: two patients had mild HA. Theere were 3 patients with HA demonstrated in Table 1.

We revised this in the current version of the paper. It was three patients with HA. 

5. Grammer

Puncture of the femoral vein was performed under ultrasound "guiding".

Puncture of the femoral vein was performed under ultrasound "guidance".

This grammer typo was changed to "guidance".

Reviewer 2 Report

This paper presents a widely used ablation procedure in a group of very rare patients with bleeding disorders. The work is very interesting and will facilitate the care of patients after ablation procedure by using appropriate medications adapted to their bleeding disorders.

Minor remarks for Authors

1. In the "3. Results" section, you first mention 7 patients, and in the next sentence, when allocating patients to specific blleding disorders, you only name 6 of them.

2. Tables 2 and 3 require aesthetic refinement. Changing the font so that the words fit on one row and widening some columns would make the tables easier to read.

3. The text requires some editorial correction. There are unnecessary spaces in the text, for example in lines 208 and 226.

4. The presence of original figures of the results of patient examinations, for example ablation, would enrich the aesthetic value of the article.

Author Response

This paper presents a widely used ablation procedure in a group of very rare patients with bleeding disorders. The work is very interesting and will facilitate the care of patients after ablation procedure by using appropriate medications adapted to their bleeding disorders.

Thank you for this valuable comment.

Minor remarks for Authors

  1. In the "3. Results" section, you first mention 7 patients, and in the next sentence, when allocating patients to specific bleeding disorders, you only name 6 of them.Thank you for this comment. We revised this typo in the revision of the paper. "Three patients had VWD Type I, one had VWD Type IIa. Three patients had mild HA."

2. Tables 2 and 3 require aesthetic refinement. Changing the font so that the words fit on one row and widening some columns would make the tables easier to read.

Thank you. We performed an refinement of the tables.

3. The text requires some editorial correction. There are unnecessary spaces in the text, for example in lines 208 and 226.

Thank you. We performed the editorial corrections.

4. The presence of original figures of the results of patient examinations, for example ablation, would enrich the aesthetic value of the article.

Thank you. A figure of an PVI was addedd to the paper.

Reviewer 3 Report

What the findings are, and then possibly indicate its strengths? What is the main question addressed by the research? Is it relevant and interesting? The article concerns the safety of invasive treatment in patients with coagulation disorders - the observation included 7 patients, 4 patients with VWD and 3 patients with HA (mild form). Patients with coagulation disorders are patients with an increased risk of bleeding complications. However, the observed group belonged to the group with a moderate risk of complications, especially patients with HA, in whom the level of factor VIII practically allowed for a fairly safe incidental procedure. Secondly, it should be mentioned that endovascular procedures are procedures with a low risk of bleeding complications, especially in the venous area, including procedures and patients with arrhythmias. In the work, the authors confidently presented a scheme aimed at increasing safety: 1 / all patients were treated with substitutions in the period before the procedure, which allowed for safe use of heparin and the procedure. 2 / in one-year follow-up confirmed the possibility of using OAC, finding bleeding complications in only one patient. And this is the most interesting aspect of the work, which concerns the possibility of treating OAC in patients with coagulation disorders. However, it should be noted that the observation concerns only 7 patients. How original is the topic? What does it add to the subject area compared with other published material? The number of reports on the treatment of OAC in patients with coagulation disorders is not large. The works mainly concern the safety of surgical treatment in patients with severe coagulation disorders (HA <1% VIII), as numerous haemorrhagic complications, often life-threatening. Is the paper well written? Yes Is the text clear and easy to read? 

Yes.

Author Response

Thank you for your valuable comments!

Reviewer 4 Report

The number of patients in the study is very small due to the incidence of the disease. Also I am not sure how this is important to readers as hematoma after venous access is not really an issue in clinical practice.

Were all wVD patients "symptomatic"? You mentioned most of these patients are asymptomatic in the introduction line 43.

Line 187, no complications is an overstatement. You had one hematoma, which is a minor complication.

Line 192, I wonder if your complication rate low because of the use of the factors administered prior to the procedure or because the authors used ultrasound. I personally have not had hematoma complications after use of 24Fr sheath (PE thrombectomy cases) in femoral veins on patients with continuous heparin drip. Cardiologists, especially EP doctors are notorious for no ultrasound puncture because it is a "venous" access. What is the complication rate in your institution for non vWD/HA patients?

Lines 203-209 should be in methods section as these are your protocol used in the study patients.

Author Response

The number of patients in the study is very small due to the incidence of the disease. Also I am not sure how this is important to readers as hematoma after venous access is not really an issue in clinical practice.

Thank you very much for this important comment. In our expirience an hematome of the groin can lead to massively consequences to the patient including surgery repair of the venous bleeding. Therefore, in our opinion the topic is very relevant for our patients.

Were all wVD patients "symptomatic"? You mentioned most of these patients are asymptomatic in the introduction line 43.

Thank you for this question. In fact 3 patients with HA were asymtpmatic and 2/4 patients with vWD were asymptomatic.

Line 187, no complications is an overstatement. You had one hematoma, which is a minor complication.

Thank you. We changed the term to "major" only. 

Line 192, I wonder if your complication rate low because of the use of the factors administered prior to the procedure or because the authors used ultrasound. I personally have not had hematoma complications after use of 24Fr sheath (PE thrombectomy cases) in femoral veins on patients with continuous heparin drip. Cardiologists, especially EP doctors are notorious for no ultrasound puncture because it is a "venous" access. What is the complication rate in your institution for non vWD/HA patients?

Thank you very much. The rate of major bleeding of the groin is about 5% in our institution.

Lines 203-209 should be in methods section as these are your protocol used in the study patients.

Thank you! We revised this in the current version.

Round 2

Reviewer 1 Report

No further comments.

Author Response

Thank you for this favorable comment

Reviewer 4 Report

Thank you for answering my questions. I suggest adding your institutional experience of 5% complication in non-vWD/HA after the ablation procedure in the discussion so that we can see your results better with your ultrasound-guided technique.

Author Response

Thank you for answering my questions. I suggest adding your institutional experience of 5% complication in non-vWD/HA after the ablation procedure in the discussion so that we can see your results better with your ultrasound-guided technique.

Thank you for this comment. The number wad added to the latest version of the revised paper. Please see lines 193 + 194.